# New Intravenous Calcimimetic Agents: New Options, New Problems. An Example on How Clinical, Economical and Ethical Considerations Affect Choice of Treatment

**DOI:** 10.3390/ijerph17041238

**Published:** 2020-02-14

**Authors:** Giorgina Barbara Piccoli, Tiziana Trabace, Antoine Chatrenet, Carlos Alberto Carranza de La Torre, Lurlinys Gendrot, Louise Nielsen, Antioco Fois, Giulia Santagati, Patrick Saulnier, Nicola Panocchia

**Affiliations:** 1Department of Clinical and Biological Sciences, University of Torino, 10124 Torino, Italy; 2Nephrologie, Centre Hospitalier Le Mans, 72037 Le Mans, France; tizi.trb87@gmail.com (T.T.); antoine.chatrenet@gmail.com (A.C.); just752@hotmail.com (C.A.C.d.L.T.); lgendrot@ch-lemans.fr (L.G.); lnielsen@ch-lemans.fr (L.N.); afois@ch-lemans.fr (A.F.); giulia.santagati@hotmail.it (G.S.); 3Statistical laboratory, University of Angers, 49035 Angers, France; patrick.saulnier@chu-angers.fr; 4Nephrology Department, Fondazione Policlinico Universitario A. Gemelli, IRCCS, 00168 Roma, Italy; nicola.panocchia@policlinicogemelli.it

**Keywords:** principlist ethics, drug choice, hemodialysis, calcimimetic agents, non-compliance, economic aspects of drug choice

## Abstract

Background. Dialysis treatment is improving, but several long-term problems remain unsolved, including metabolic bone disease linked to chronic kidney disease (CKD-MBD). The availability of new, efficacious but expensive drugs (intravenous calcimimetic agents) poses ethical problems, especially in the setting of budget limitations. Methods. Reasons of choice, side effects, biochemical trends were discussed in a cohort of 15 patients (13% of the dialysis population) who stared treatment with intravenous calcimimetics in a single center. All patients had previously been treated with oral calcimimetic agents; dialysis efficacy was at target in 14/15; hemodiafiltration was employed in 10/15. Median Charlson Comorbidity Index was 8. The indications were discussed according to the principlist ethics (beneficience, non maleficience, justice and autonomy). Biochemical results were analyzed to support the clinical-ethical choices. Results. In the context of a strict clinical and biochemical surveillance, the lack of side effects ensured “non-maleficence”; efficacy was at least similar to oral calcimimetic agents, but tolerance was better. Autonomy was respected through a shared decision-making model; all patients appreciated the reduction of the drug burden, and most acknowledged better control of their biochemical data. The ethical conflict resides in the balance between the clinical “beneficience, non-maleficience” advantage and “justice” (economic impact of treatment, potentially in attrition with other resources, since the drug is expensive and included in the dialysis bundle). The dilemma is more relevant when a patient’s life expectancy is short (economic impact without clear clinical advantages), or when non-compliance is an issue (unclear advantage if the whole treatment is not correctly taken). Conclusions. In a context of person-centered medicine, autonomy, beneficence and non-maleficence should weight more than economic justice. While ethical discussions are not aimed at finding “the right answer” but asking “the right questions”, this example can raise awareness of the importance of including an ethical analysis in the choice of “economically relevant” drugs.

## 1. Introduction

Dialysis treatment is continuously improving, ensuring better tolerance and overall higher survival of a population that is steadily getting older and more complicated. Although the therapeutic armamentarium is now far more varied than in the past, virtually all the long-term problems related to uremia and its treatments are still unsolved: cardiovascular diseases are still the main cause of death in a context of accelerated vascular ageing; vascular access malfunction still makes it often difficult to optimize treatment; Chronic Kidney Disease related Mineral and Bone Disorder (CKD-MBD) remains one of the plagues of long-term dialysis treatment [1,2,3,4,5,6].

The focus on CKD-MBD has changed, and attention has progressively shifted from the effects it has on bones (osteopenia, pathologic fractures, reabsorptions, geodes, etc.) to its effects on the cardiovascular system (vascular calcifications and decreased myocardial contractility; myocardial hypertrophy and fibrosis; valvular, myocardial and arterial calcifications) [7,8,9].

While it is impossible to reduce the effect of the entangled derangements of such a complex system to one element only, the correction of hyperparathyroidism is still an unmet challenge, in which many questions have remained unanswered. Parathyroid hormone (PTH) is a uremic toxin, high levels of which are associated not only with the classic hallmarks of bone disease, and are characterized by an increase in calcium, phosphate and FGF-23, all of them associated with an increased mortality, but also with anemia, hypertension, neuromyopathy, dysfunction of the peripheral and central nervous systems, and various metabolic effects, including dyslipidemia and impaired insulin secretion [10,11,12,13,14,15,16,17,18,19,20,21,22].

What our target for PTH should be is still a matter of controversy. Many studies have found an inverse relationship between serum PTH levels and mortality in dialysis patients, especially when PTH levels are above 300 pg/mL [23,24,25,26,27,28]. Different medical societies have proposed different goals, ranging from the target set by the Japanese Society of Nephrology (60–240 pg/mL), up to the recent KDIGO (Kidney Disease Improving Global Outcomes) guidelines (up to 9 times the upper limit of normality) [23,27,28,29,30,31].

The availability of calcimimetic agents, first oral and presently also intravenous, was therefore a major advance, as the recent KDIGO Clinical Practice Guidelines Update for the Diagnosis, Evaluation, Prevention, and Treatment of Chronic Kidney Disease-Mineral and Bone Disorder (CKD-MBD) clearly acknowledged [29,30,31].

Calcimimetic agents are expensive, however. At a time of global economic crisis, in which the burden of end-stage chronic kidney disease (ESRD) is critical, cost is an important factor in deciding what treatment to choose. In fact, to the best of our knowledge, the NICE (National Institute fro health and Care Excellence) guidelines on the use of Etelcalcetide in CKD-MDB are the first to explicitly raise the question of cost reduction as a point that needs to be taken into consideration in choosing a calcimimetic agent for IV use [32].

While this pragmatic approach may be seen as ethically disturbing since it heavily impacts on a clinical choice, it also underlines the potential strength of scientific societies in the discussion on treatment costs and availability, which in turn influence clinical choices [32,33,34,35,36,37].

The way dialysis is reimbursed plays an important role, since the intravenous drugs administered during the dialysis session are usually part of the dialysis budget (dialysis “bundle”) and previous experience with erythropoietin clearly shows that the method of reimbursement deeply affects clinical practice [34,35,36,37,38,39]. Conversely, oral drugs are usually outside the dialysis budget, and this is frequently the case for first generation oral calcimimetic agents. While the price of an average dose of oral calcimimetic and phosphate binders may be equivalent to that of second-generation IV calcimetic agents, the schizophrenia intrinsic in a multi-payer system often discourages use of the latter agents, if they are included in the dialysis bundle [34].

The life expectancy of a patient with multiple comorbidity is often short. The patient’s treatment compliance is likely to be low, and often without a clear histological staging. Grading a disease is not an easy task and prescribing a new drug, particularly in the context of a multi-drug regimen cannot be based on simplistic models [40,41].

In an era of “personalized medicine”, the clinical issues, influenced by economic ones, merge with ethical ones. Health technology assessment refers to the systematic evaluation of properties, effects, and/or impacts of health technology. It is a multidisciplinary process to evaluate the social, economic, organizational and ethical issues of a health intervention or health technology. Questions like “What should our treatment targets be?” or “Is it worth employing an expensive treatment in a patient with a low life expectancy?” and further “Is it logical to invest in patients with low compliance with overall treatments?” do not have univocal answers.

The present case series is therefore discussed as a means to focus attention on the interactions between the clinical, ethical and economic issues involved in the prescription of a new intravenous calcimimetic agent and underline the need for extending ethical considerations to encompass the daily choices faced in clinical practice.

## 2. Materials and Methods

### 2.1. Setting and Patients

The setting of the study is the Centre Hospitalier du Mans, which is presently one of the three largest non-university hospitals in France. The pool of patients on chronic hemodialysis–hemodiafiltration ranges from 95 to 110, depending on the incidence of kidney transplantation, death and transfers. In keeping with the indications of the French Society of Nephrology (Société Francophone de Néphrologie, Dialyse et Transplantation, SFNDT [42]), the population studied is a large sample of patients at high comorbidity. At the time of the study, about 70% of the patients were being treated with on-line hemodiafiltration (HDF), while the rest were being treated with conventional hemodilaysis (HD). The dialysate is acetate-free and citrate-based (calcium concentration 1.5–1.75 mmol/L; sodium 138–140 mEq/L, bicarbonate 32–36, temperature 36–37 °C; potassium: 2–3mmol/L). HDF employs high permeability membranes, with different surfaces; post-dilution HDF is employed in about half of the cases, the rest being treated with pre-dilutional HDF [43,44]. Conventional hemodialysis is performed with the same dialysate, employing medium-low permeability dialyzers. About two thirds of the patients were dialyzed with an arteriovenous fistula (AV fistula), the rest with a permanent tunnelized catheter. The dialysis policy is aimed at personalization of treatment, with incremental and more-frequent dialysis [43,44].

Dialysis efficiency was calculated using Daugirdas II Kt/V and comorbidity was assessed using the Charlson Comorbidity Index (CCI). Subjective Global Assessment (SGA) and Malnutrition Inflammation Score (MIS) were also calculated in all cases [43,44,45]. Beta-2 microglobulin was assessed 6 times per year.

### 2.2. Selection Criteria and Treatment Evaluation

Patients who were started on Etelcalcetide between January 2018 and January 2019 were included in the present descriptive, retrospective study, aimed at obtaining a narrative analysis of the indications and contraindications to treatment from a clinical and ethical point of view.

The indications were reviewed and discussed collegially; patients were monitored at least each 2 weeks until stabilization, and then at least monthly. Monthly tests included urea, creatinine, albumin, total proteins, Na, K, Ca, phosphorous, HCO3, at start and end of the dialysis session; vitamin D, parathyroid hormone (PTH), blood cell count, ferritin, C-reactive protein and brain natriuretic peptide. Other tests were performed with personalized frequency.

Side effects were recorded in the clinical charts, and were reviewed collegially for the present study.

### 2.3. Statistical Analysis

A descriptive statistical analysis was employed to describe and contextualize data with respect to our dialysis ward, and to allow comparisons with other settings. Descriptive analysis was expressed as mean ± standard deviation (SD) or median and inter-quartile range (IQR) when appropriate. Comparisons between groups were assessed by *t*-test for paired or unpaired data.

Trend analysis was limited to visual appreciation of the pattern observed over time; correlation between parathyroid hormone and phosphate levels was performed employing a linear model.

While an efficacy analysis is beyond the scope of the present report, dealing with a small number of heterogeneous cases, data were gathered and analyzed at different time points, before and during treatment.

### 2.4. Ethical Discussion

The ethical discussion of the clinical indications follows the four principles of principlist ethics (beneficence, non-maleficence, autonomy and justice) [46,47,48].

For the sake of the present discussion, given the importance of costs, we considered justice as synonymous with distributive justice (the use of expensive treatments should be limited in settings with budget constraints) [48,49]. Data were discussed collegially and the conclusions were summarized narratively.

### 2.5. Ethical Issues

All the patients included were adults and agreed to allow use of their routine clinical and biochemical data for the sake of this study. All patients signed a written consent for analysis and publication of their data in anonymous form.

The study did not involve additional blood tests or additional imaging techniques and was approved by the hospital’s ethics committee (“Avis favorable du groupe d’éthique du Centre Hospitalier du Mans du 12/06/2018”).

## 3. Results

### 3.1. Overall Data

Overall data on the 15 patients who started at least a period of treatment with Etelcalcetide (prevalence 13% of the current dialysis population) are summarized in Table 1.

All these patients had previously been treated with oral calcimimetic agents. In all cases, the previous schedules were not able to ensure satisfactory control of PTH or hypercalcemia (see below).

At the start of treatment with intravenous calcimimetic drugs, dialysis efficacy was at target in all but one patient, who had vascular access problems; most of the patients were on HDF, with highly permeable membranes. Beta 2 microglobulin was in an acceptable range in most cases. When high, its levels were considered to be linked to a disease condition rather than to underdialysis.

Charlson Index in 5 patients was at or below 6, the limit usually employed to discriminate high comorbidity, and was above 9 in 3 (very low life expectancy).

In all patients, we observed significant variability in all measures (Figure 1).

### 3.2. Overall Indications for Treatment

Table 2 summarizes the main clinical indications that were considered and balanced in the prescription of intravenous calcimimetic agents.

Insufficient efficacy of oral calcimimetic agents in PTH control was the main, and in some cases the only indication, for switching from oral to intravenous calcimimetic agents. The lack of increase in alkaline phosphatses was a controversial issue: We felt that trying to lower exceedingly high PTH levels could be of interest considering the pleiotropic effect of the hormone, as described above. The fact that lowering PTH allowed lowering phosphate levels (as described below) led us to continue the treatment, as it suggested that bone reabsorption was indeed an issue in these patients.

In only one case, the main goal was hypercalcemia, in a setting of normal, but non-suppressed PTH level; this indication was combined with nausea and low tolerance to oral calcimimetics. This patient had two reasons for being hypercalcemic: systemic tuberculosis and a small parathyroid adenoma. Calcimimetic treatment was suggested by the consultant endocrinologist on the account of the need for correcting hypercalcemia (contraindications to steroids; parathyroidectomy not accepted by the patient; unsuppressed PTH levels) in the wait for response to antitubercular treatment. Antitubercular treatment allowed correction of hypercalcemia; frank hyperparathyroidism (PTH 400–800 pg/mL) became evident and the patient is now scheduled for adenomectomy. This is the only patient who discontinued treatment, due to persistence of nausea and vomiting, in a setting of oral treatment of tuberculosis.

Tolerance is frequently a related issue, since most patients do not tolerate high doses of oral calcimimetic agents, and, interestingly, IV treatment was less associated with nausea or gastrointestinal problems, which in our series were only reported by two patients, one of whom discontinued treatment.

The second most frequent indication was poor adherence to oral calcimimetic agents. This was the only indication in three patients and was associated with nutritional problems in one further patient. Other indications and combinations are discussed in the following paragraphs (Table 2).

### 3.3. Ethical Issues: Insufficient Efficacy with or without Low Tolerance to Oral Calcimimetic Agents

This subset of patients includes three patients with long dialysis vintage, severe CKD-MBD and dialysis-related amyloidosis, and four older patients at high comorbidity: all suffer from severe peripheral vascular disease and two have undergone toe or limb amputation.

In all these cases, the lack of side effects ensures “non-maleficence”. “Autonomy”, also in patients with high score comorbidity autonomy was respected through the involvement of patients according to the shared decision-making model and confirmed by and good adherence to the treatments. In this setting, the ethical issue resides in the balance between the clinical “benefit” in patients with either short life expectancy and/or well-established bone or vascular disease and “justice” (i.e., the economic impact of treatment choice). The dilemma is more relevant when a patient’s life expectancy is short, as shown by a high Charlson Comorbidity Index (Table 3).

### 3.4. Ethical Issues: Non Compliance

Non-compliance to oral drugs was the main reason for prescribing intravenous calcimimetic agents for four patients (Table 3). For two of them, non-compliance was associated with other issues (diffuse, severe vascular disease in a young patient, and oral absorption in a severely vasculopathic patient on enteral nutrition via gastrostomy).

In these cases, non-maleficence and autonomy are not relevant, since the patients agreed to intravenous treatment, and eventually appreciated the reduced pill burden; no side effects were found, as long as calcium levels were closely monitored in the patient with borderline-low calcium levels at the start of treatment, due to non-compliance to oral calcium.

The main issue was deciding whether to employ an expensive drug to “simplify” the management of a patient who was not participating actively in his/her own treatment. Lack of compliance can make it impossible to reach the targets (calcium, phosphate) which combine with PTH to maximize benefits. Once more, the benefit of cardiovascular health is not clear in patients who are already heavily calcified, or have experienced severe cardiovascular morbidity (Broca aphasia, due to cerebral ischemia; distal amputation in the context of a diabetic foot).

### 3.5. Ethical Issues: Other Complex Indications

The expected advantage of reducing PTH or calcium levels in the context of hypercalcemia, in combination with mild hyper PTH and low tolerance to oral calcimimetic drugs has to be balanced with complex clinical situations, in which, for example, hypercalcemia is associated with a granulomatous disease. In these cases, treatment with an intravenous calcimimetic agent may be seen as a last-ditch attempt and considered “too expensive”, given that there is no guarantee that the patient will benefit from the change.

Once more, uncertainty in the efficacy of the treatment in the long-term prognosis and costliness were the relevant issues, while non-maleficence and autonomy were not, as the patients in the study had agreed to intravenous treatment, and no side effects were found.

### 3.6. Clinical Results: Why Treatment Is Being Continued

Figure 2 reports the patterns of PTH and calcium and phosphate levels observed in patients with at least 6 months of treatment. The high variability of responses and oscillations in the biological markers is evident, and the need for strict controls has to be stressed, also for allowing identification of the treatment response (Figure 3a,b).

Overall the treatment lowered PTH levels, the exceptions being patients who started treatment in the context of hypercalcemia (Figure 4).

## 4. Discussion

Twenty-five years ago, in 1995, an editorial entitled “Who owns medical technology” appeared in The Lancet. It described the impact of new technologies on moral values, with the example of the introduction of a snowmobile in a Lapp community [50]. The new technology impacted daily life and ultimately on the moral values of a previously almost autarchic community. Technology brings more than hardware, was perhaps the obvious, but not simplistic conclusion of the paper [50].

The issue discussed here may be seen as an example of the influence of new technologies, including drug development, on the ethical framework in which clinicians work. In this regard, the case here discussed is not exceptional and may be seen as an example of the need for integrating economic and ethical issues in the routine clinical reasoning.

A new-generation drug (intravenous calcimimetic) appears on the market, for the treatment of CKD-MBD, still an issue in dialysis patients. The first randomized controlled trials are favorable in terms of safety and efficacy [51,52,53,54,55]. Even considering the potential interference of the pharmaceutical industry in enhancing positive reports, the drug seems a major advance in the care of a challenging clinical problem [51,52,53,54,55,56,57,58,59,60,61,62,63,64]. Pharmacokinetic studies, animal data and expert opinions indicate that the new agent has obvious advantages over the oral drug in improving compliance (IV on dialysis instead of orally once or twice daily); it seems to have a better efficacy profile and appears to increase stability in the blood levels of calcium, phosphate and PTH, in keeping with a longer duration of action and lack of interference with food intake; its side effects are similar to those caused by earlier products, and can generally be avoided with a strict control policy [65,66,67,68,69,70,71,72,73,74,75,76,77,78,79,80,81,82,83,84].

However, the evidence is still limited and the new drug is expensive. In fact, it is deemed so expensive that, to the best of our knowledge, for the first time in renal medicine, a reduction in price is cited as a requisite for its use in a reference guideline [32,85,86,87,88,89].

While a commentary on the schizophrenia intrinsic in many payment systems is beyond the scope of the present discussion, we wish to note that cost differences are also determined by differences in providers: at least in many European settings, drugs linked to dialysis sessions are calculated as part of an overall reimbursement “bundle” per dialysis sessions, while oral drugs are not. Thus, cost differences between oral (not included in the hospital’s budget) and intravenous (included in the hospital’s budget) drugs may be perceived by the provider (hospital) as too high to be acceptable, even if the global difference is low [32,33].

This is probably necessary, but for many physicians, a quite disturbing influence by an economic aspect on a medical choice, which leads to a series of questions, that impact on prescription. In this paper, we have based our analysis of the impact of ethical issues on the clinical choice of whether or not to use a new intravenous calcimimetic agent, on the four principles of principlism [46,47,48].

The first question is efficacy (beneficence): the present PTH target range is broad, and within it, personalized targets vary depending on the importance attributed to controlling phosphate levels, which are associated with PTH levels. In patients with long life expectancy, prevention of vascular disease generally is crucial and PTH levels should probably be kept in the lower target range to minimize vascular damage. Conversely, higher PTH levels may be acceptable in very old patients and in the presence of low bone turnover, shown by low alkaline phosphate levels [23,24,25,26].

These clinical approaches raise an ethical question: in the lack of a clearly demonstrated advantage of zealous (and possibly overzealous) correction of PTH levels in our population, should physicians opt for a possibly useless (but harmless) excess of care or risk under-treatment? In other words, should we insist on correcting PTH levels in patients whose life expectancy is short or where end-organ damage (diffuse vascular calcifications or severe osteopenia) is already present?

In the absence of a sound clinical answer, the choice may follow the ethical principle of “beneficium”, maximizing a potentially useful treatment versus the choice of limiting expensive therapies to patients who have a clear benefit. If we focus on benefit, our choice must be to do as much as we can for every patient, in the absence of side effects, and this is what we chose to do in our center. The answer may be different if we focus on justice, seen in economic terms; in this case, the option would be not to use the new medications to treat patients with a very low life expectancy, or established vascular and bone diseases, which are unlikely to be reversible.

Person-centered medicine could merge the two options: this approach includes a reduction in the burden of drugs and adaptations of protocols, such as relaxation of biochemical targets (i.e., more permissive hyperphosphatemia and hyperparathyroidism in patients with low life expectancy). According to this approach, autonomy (pre-eminence of the patient’s values, in particular his/her quality of life), beneficence (advantages of short-term treatment), non-maleficence (burden of treatment) weight more than economic justice in leading choices [90].

Likewise, is it reasonable to use expensive treatments for patients whose low compliance may impair reaching the targets (calcium and phosphate levels), necessary for attaining the effect of the chosen treatments, or is it correct to limit their use to treating compliant patients?

Similar consideration was discussed concerning the care of obese patients and re-transplantation in subjects who had lost a previous graft due to non-adherence to treatments [91,92,93,94,95,96].

We chose to be our patients’ advocate, and to try to maximize potential benefit, regardless of life expectancy, and of the presence of established cardiovascular and bone disease. Furthermore, we chose to use lower targets in patients with severe vascular disease, thus putting beneficence and non-maleficence first, with respect to justice (and economy). While we were fortunate to work in a setting in which economic considerations were not the only criteria for selecting treatments, we consider that a clear appraisal of ethical issues is a useful tool in discussing therapeutic choices.

Although efficacy was not the principal goal of our study, given the small group of patients involved and the variability of their biological data (Figure 1, Figure 2, Figure 3 and Figure 4), our analysis suggests that the policy adopted was harmless, and may have improved overall control of PTH. In addition, it was at least as efficacious as the previous one, with better tolerance.

Better appraisal of benefits, possible with larger multicentric analyses, is needed to confirm the positive results, at least in terms of compliance and patients’ appreciation. A close biochemical monitoring is fundamental to be able to correctly identify trends. While economic issue, national or local guidelines often limit the clinical choices, in our setting expensive choices of new drugs are allowed, but they must be motivated.

This narrative ethical analysis has the limit of being monocentric and of regarding a single drug and a small number of cases, but has the advantage of using a novel approach to the analysis of treatment choices, trying to raise physicians’ and policymakers’ awareness of the ethical aspects of apparently “technical “ decisions.

## 5. Conclusions

The case discussed above is an example of how economic considerations have an impact on what treatment should be chosen. Would the choice between an oral or intravenous calcimimetic agent be different if their costs were equivalent? If the answer is yes, then we have to acknowledge that the cost issue, reflected in the ethical predominance of “economic” justice, is the leading principle. Conversely, a choice based on clinical issues, putting any possible advantage first, in the absence of harm, and with the patient’s agreement, with respect to economic issues, represents different ethical priorities.

The aim of ethical discussions is not to find “the right answer” but instead to ask “the right questions”. In this setting, we consider that this example can raise awareness of the importance of taking into account not only the clinical and economic aspects, but also the ethical ones, involved in the choice of “economically relevant” drugs.

## Figures and Tables

**Figure 1 ijerph-17-01238-f001:**
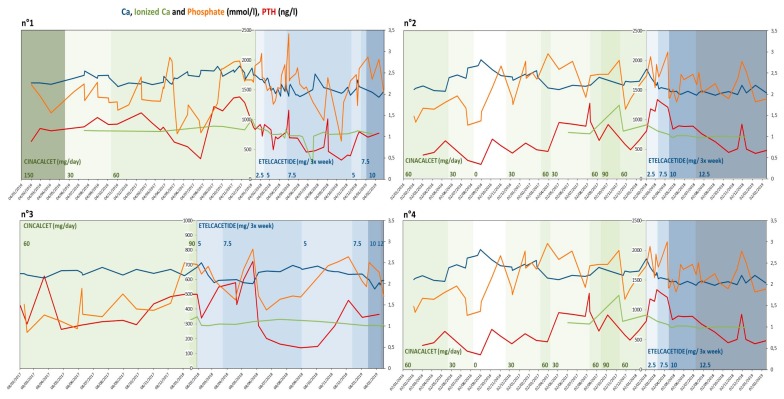
Variability of PTH, calcium and phosphate levels in the four patients with at least one year of follow-up. Legend: Ca Calcium; PTH parathyroid hormone. N: patient number.

**Figure 2 ijerph-17-01238-f002:**
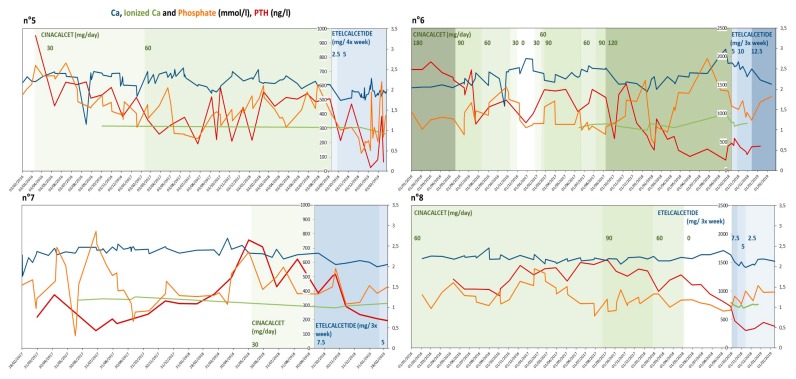
PTH, calcium and phosphate levels in patients with 5–6 months of follow-up. Legend: Ca calcium; PTH parathyroid hormone. N patient number.

**Figure 3 ijerph-17-01238-f003:**
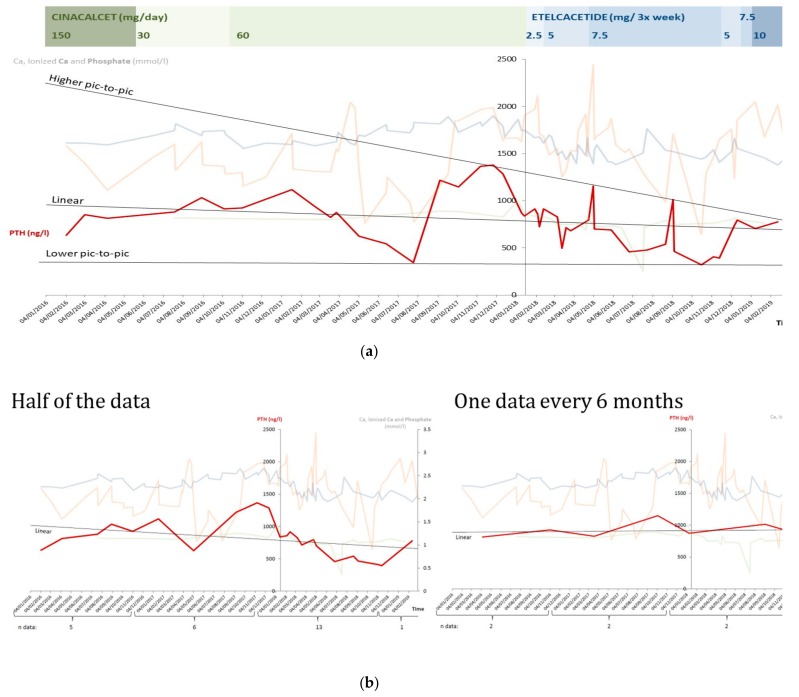
Different trends of PTH in the same case, according to the modality of analysis. Legend: Ca calcium; PTH parathyroid hormone. (**a**): all PTH data available; (**b**): limited PTH data available.

**Figure 4 ijerph-17-01238-f004:**
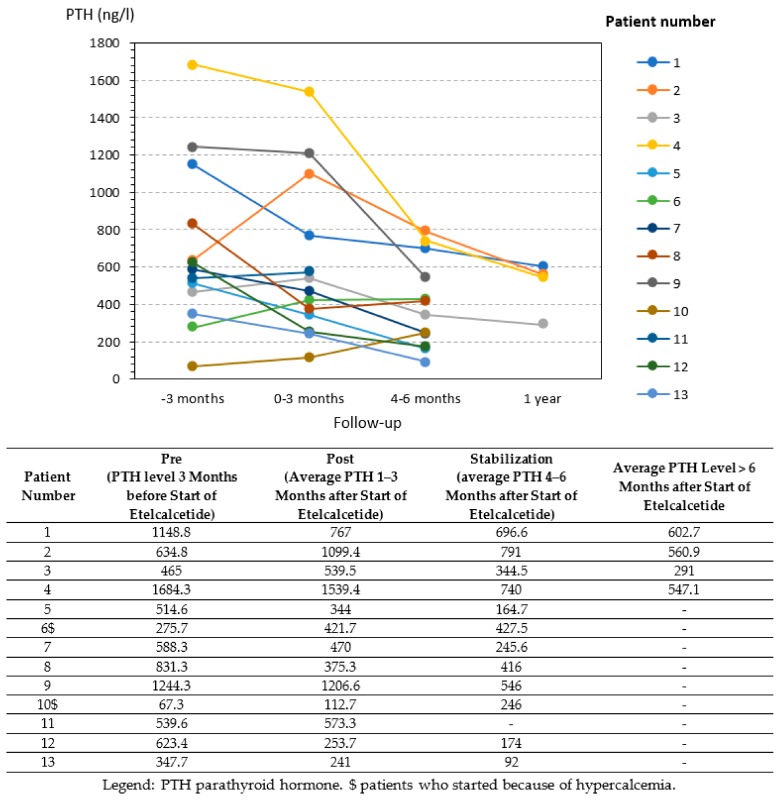
Parathyroid hormone trends in the study group.

**Table 1 ijerph-17-01238-t001:** Baseline clinical data in the selected population.

Patient	Age	Sex	Start	CCI	Dialysis Vintage (years)	Main Indication for Etelcalcetide	Dialysis Schedule	Kt/V	Beta2M	Metabolic Bone Disease
1	78	F	01/2018	6	8	Low tolerance, insufficient PTH control	HDF-Pre 4 h × 3	1.5	26.4	Osteopenia
2	82	F	02/2018	8	9	Pathologic fractures, insufficient PTH control	HD 4 h × 3	1.9	27.4	Pathologic fractures
3	57	M	02/2018	3	15	Insufficient PTH control, long RRT vintage	HDF post 4 h × 3	1.79	33.5	B2M amyloidosis
4	83	F	02/2018	10	12	Insufficient PTH control	HDF post 3 h 30 × 3	1.59	23.5	Osteopenia
5	79	F	09/2018	10	7	Insufficient PTH control, severe vascular disease	HDF Pre 3 h 30 × 4	1.50	18.6	Osteopenia
6	52	M	10/2018	6	11	Need for high calcimimetic dose	HD 3 h 30 × 3	1.24	33.1	B2M amyloidosis, Osteopenia, deformities
7	59	F	10/2018	9	2	Absorption problems, low tolerance, low compliance	HDF post 4 h × 3	1.30	36.6	-
8	35	M	10/2018	4	6	Low compliance	HDF post 4 h × 3	1.33	29.6	-
9	37	F	11/2018	11	7	Irregular compliance	HDF pre 4 h × 3	1.71	61.3	-
10	54	F	11/2018 to ** 02/2019	4	1	Hypercalcemia, low tolerance	HD 4 h × 3	1.51	26.2	-
11	70	F	11/2018to * 23/2018	9	15	Mild hypercalcemia, amyloidosis	HD 4 h × 3	1.6	14.5	B2M amyloidosis, osteopenia
12	43	M	11/2018	6	1	Low compliance, diffuse calcifications	HD 4 h × 3	1.08	16.1	-
13	63	M	11/2018	7	20	Insufficient PTH control, long RRT vintage	HDF post 4 h × 3	1.41	24.1	B2M amyloidosis
14	83	F	02/2019	9	10	Low compliance, severe vascular disease	HDF pre 4 h × 3	1.32	31.1	Osteopenia
15	67	F	02/2019	9	40	Insufficient PTH control, long RRT vintage	HDF Mid	1.52	21.6	B2M amyloidosis, osteopenia
Median	63	10 F		8	9			1.51	26.8	
Min	35	5 M		3	1			1.08	14.5	
Max	83			11	40			1.90	61.3	

Note: data recorded at the start of treatment. Legend: TTT: Treatment. CCI: Charlson Comorbidity Index. HD: hemodialysis. HDF: hemodiafiltration. Pre: predilutional; post: post dilutional; mid: mid dilutional. PTH: parathyroid hormone. RRT: renal replacement therapy. Beta2M: Beta 2 Microglobulin. Kt/V: assessed via Daugirdas 2. H: hours. M male; F female. * psychological reasons: preference not to use “new drugs”. ** low tolerance: present also with Cinaclacet, nausea persists after discontinuation (on anti tuberculosis treatment).

**Table 2 ijerph-17-01238-t002:** Pros and cons of treatment: a clinical approach.

Patient	Age	Main Indication for Etelcalcetide	PROS	CONS
1	78	Low tolerance, insufficient PTH control	Interest in ameliorating metabolic control in an elderly woman with relatively long vintage, high fracture risk and diffuse vascular calcifications.	Reversing bone damage is probably difficult, short life expectancy. No increase in alkaline phosphatase, possible low-turnover bone disease.
2	82	Pathologic fractures, insufficient PTH control	Interest in reducing risk of fractures in an elderly woman with relatively long vintage, previous fractures and diffuse vascular calcifications.	Reversing bone damage is probably difficult, short life expectancy. No increase in alkaline phosphatase, possible low-turnover bone disease.
3	57	Insufficient PTH control, Long RRT vintage	Interest in reducing all the long- term effects of dialysis in a young patient with diffuse vascular calcifications and low transplant chances (hyper immunized).	Reversing bone damage is probably difficult. No clear relationship with Beta2 amyloidosis.
4	83	Insufficient PTH control	Interest in ameliorating metabolic control in an elderly woman with relatively long vintage, high fracture risk and diffuse vascular calcifications.	Reversing bone damage is probably difficult, very short life expectancy. No increase in alkaline phosphatase, possible low-turnover bone disease.
5	79	Severe vascular calcifications, insufficient PTH control	Interest in ameliorating metabolic control in an elderly woman with relatively long vintage, severe and diffuse vascular calcifications.	Reversing bone damage is difficult, very short life expectancy. No increase in alkaline phosphatase, probably a low-turnover bone disease. No demonstration of efficacy in retarding vascular disease, if Ca-P balance is acceptable.
6	52	Need for high calcimimetic dose, bone fractures and deformities. Severe hypercalcemia	Interest in reducing risk of fractures in a young patient with a very long RRT vintage, previous fractures, deformities and diffuse vascular calcifications.	Reversing bone damage is probably difficult. No increase in alkaline phosphatase, probably a low-turnover bone disease. No demonstration of efficacy in retarding vascular disease, if Ca-P balance is acceptable.
7	59	Absorption problems, low tolerance, low compliance	Interest in ameliorating metabolic control in a relatively young woman at very high risk for vascular events (aphasic and with motor deficit after a cerebral accident). Reducing pill burden may improve compliance for other drugs.	No demonstration of efficacy in retarding vascular disease, if Ca-P balance is acceptable. Low compliance may offset the advantages of the drug, in particular in the presence of hyperphosphatemia.
8	35	Low compliance	Interest in ameliorating metabolic control in a young man with low compliance. Reducing pill burden may improve compliance for other drugs.	Low compliance may offset the advantages of the drug, in particular in the presence of hyperphosphatemia.
9	37	Irregular compliance	Interest in ameliorating metabolic control in a young woman with very high comorbidity. Reducing pill burden may improve compliance for other drugs.	No demonstration of efficacy in retarding vascular disease, if Ca-P balance is acceptable. Low compliance may offset treatment advantages.
10	54	Hypercalcemia, low tolerance	Interest in ameliorating metabolic control in a relatively young woman with hypercalcemia and unadapted PTH.	The cause of hypercalcemia is probably a granulomatous disease, less prone to being improved by treatment.
11	70	Hypercalcemia, amyloidosis	Interest in ameliorating metabolic control in woman with hypercalcemia and unadapted PTH and rheumatologic disease; high comorbidity, long dialysis vintage.	Reversing bone damage is difficult, short life expectancy. No increase in alkaline phosphatase, probably a low-turnover bone disease. No clear efficacy of PTH normalization on long-term dialysis-related comorbidity.
12	43	Low compliance, diffuse calcifications	Interest in ameliorating metabolic control in a young diabetic man with severe vascular calcifications. Low compliance. Reducing pill burden may improve compliance for other drugs.	No demonstration of efficacy in retarding vascular disease, if Ca-P balance is acceptable. Low compliance may offset the advantages of the drug, in the presence of hyperphosphatemia.
13	63	Insufficient PTH control, long RRT vintage	Interest in reducing all the long-term effects of dialysis in a young patient with diffuse vascular calcifications and low transplant chances (hyper immunized).	Reversing bone damage is difficult. No clear efficacy of PTH normalization on long-term dialysis-related comorbidity, in particular B2 microglobulin deposition.
14	83	Low compliance, diffuse calcifications	Interest in ameliorating metabolic control in a young diabetic man with severe vascular calcifications. Low compliance. Reducing burden pill may improve compliance for other drugs.	No demonstration of efficacy in retarding vascular disease. Low compliance may offset the advantages of the drug, in particular in the presence of hyperphosphatemia.
15	67	Insufficient PTH control, long RRT vintage	Interest in reducing all the long-term effects of dialysis in a relatively young patient, without transplant chances (neoplasia) and with dialysis related amyloidosis.	Reversing bone damage is probably difficult. No clear efficacy of PTH normalization on long-term dialysis related comorbidity, in particular B2 microglobulin deposition.

Note: data recorded at the start of treatment. Legend: CCI: Charlson Comorbidity Index. RRT: renal replacement therapy. M male; F female. PTH parathyroid hormone.

**Table 3 ijerph-17-01238-t003:** An ethical principlist approach to the indications and contraindications to treatment.

Patient	Age	CCI	Dialysis vintage RRT	Beneficience	Non-Maleficience	Justice	Autonomy
1	78	6	8	Better PTH control; possible, but unsure reduction of vascular and fracture risk	No contra-indication	Expensive treatment without clear benefits on the main targets (bone and vascular)	Patient agreement, appreciation of the lower oral drug load.
2	82	8	9	Better PTH control; possible, but unsure reduction mainly of fracture risk	No contra-indication	Expensive treatment without clear benefits on the main targets (bone and vascular) in a patient with short life expectancy	Patient agreement
3	57	3	15	Better PTH control; possible, but unsure reduction mainly of vascular risk	No contra-indication	Expensive treatment without clear benefits on the main targets (mainly vascular)	Patient agreement, appreciation of the lower oral drug load, appreciation of metabolic improvement
4	83	10	12	Better PTH control; possible, but unsure reduction mainly of fracture risk	No contra-indication	Expensive treatment without clear benefits on the main targets (bone and vascular) in a patient with short life expectancy	Patient agreement
5	79	10	7	Better PTH control; possible, but unsure reduction mainly of vascular risk	No contra-indication	Expensive treatment without clear benefits on the main targets (mainly vascular) in a patient with short life expectancy	Patient agreement
6	52	6	11	Better PTH control; possible, but unsure reduction of vascular risk and of progression of metabolic bone disease	No contra-indication	Expensive treatment without clear benefits on the main targets (bone and vascular) in a patient with advanced end-organ damage	Patient agreement, appreciation of metabolic improvement
7	59	9	2	Better PTH control; possible reduction of vascular risk. Lack of adherence may impair reaching synergic targets (phosphate)	No contra-indication	Expensive treatment without clear benefits on the main targets (mainly vascular) in a patient with short life expectancy and in which the lack of adherence may impair reaching synergic targets	Patient agreement
8	35	4	6	Better PTH control. Possible improvement of adherence reducing the pill burden. Lack of adherence may impair reaching synergic targets	No contra-indication	Expensive treatment in a patient in which the lack of adherence may impair reaching the synergic targets (phosphate)	Patient agreement
9	37	14	7	Better PTH control. Possible improvement of adherence reducing the pill burden. Lack of adherence may impair reaching synergic targets	No contra-indication, provided mild hypocalcemia is closely monitored	Expensive treatment without clear benefits in a patient with short life expectancy and in which the lack of adherence may impair reaching the other targets combined with PTH (calcium)	Patient agreement. Appreciation of the lower oral drug load, appreciation of metabolic improvement
10	54	4	1	Hypercalcemia, low tolerance. Unclear cause of hypercalcemia, unclear whether treatment will produce improvement	No contra-indication	Expensive treatment without clear benefits in the absence of clear diagnosis.	Patient agreement
11	70	9	15	Hypercalcemia, in a context of probable moderate hyper PTH not responsive to oral calcimimetics, not well tolerated at high doses.	No contra-indication	Expensive treatment without clear benefits in the absence of clear diagnosis.	Patient agreement
12	43	6	1	Better PTH control; possible reduction of vascular risk. Possible improvement of adherence reducing the pill burden. Lack of adherence may impair reaching synergic targets	No contra-indication	Expensive treatment without clear benefits on the main targets (mainly vascular) in a patient in which the lack of adherence may impair reaching the other targets combined with PTH (phosphate)	Patient agreement. Appreciation of the lower oral drug load, appreciation of metabolic improvement
13	63	7	20	Better PTH control; possible, but unsure reduction of vascular risk	No contra-indication	Expensive treatment without clear benefits on the main targets (mainly vascular)	Patient agreement, appreciation of the lower oral drug load, appreciation of metabolic improvement
14		9	10	Better PTH control; possible reduction of vascular risk. Possible improvement of adherence reducing the pill burden. Lack of adherence may impair reaching synergic targets	No contra-indication	Expensive treatment without clear benefits on the main targets (mainly vascular) in a patient in which the lack of adherence may impair reaching the other targets combined with PTH (phosphate)	Patient agreement. Appreciation of the lower oral drug load, appreciation of metabolic improvement
15		9	40	Better PTH control; possible, but unsure reduction of vascular risk	No contra-indication	Expensive treatment without clear benefits on the main targets (mainly vascular)	Patient agreement, appreciation of the lower oral drug load, appreciation of metabolic improvement

Legend: CCI: Charlson Comorbidity Index. RRT: renal replacement therapy. M male; F female. PTH parathyroid hormone. dial vint: dialysis vintage.

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
