# Peer review of "New Intravenous Calcimimetic Agents: New Options, New Problems. An Example on How Clinical, Economical and Ethical Considerations Affect Choice of Treatment"

_ijerph, 2020, doi:10.3390/ijerph17041238_

Round 1

Reviewer 1 Report

Congratulations. 

I especially like (and I agree) with the conclusion "In a context of person-centered medicine, autonomy, beneficence, and non-maleficence should weight more than economic justice. While ethical discussions is not aimed at finding “the right answer” but asking “the right questions”, this example can raise awareness of the importance of including an ethical analysis in the choice of “economically relevant” drugs".

In any case, and beyond any intention of self-reference, there are very few papers addressing ethical considerations in Nephrology. It is for that, that I suggest you to access to our recent paper at "Clin Kidney Journal" 2020 (in press) "Evidence in CKD-MBD guidelines: is it time to treat or wait?" Bover J et al. whenever it is published. Just for the sake of sharing information. 

Reviewer 2 Report

Dear authors

I appreciate the effort to initiate a conversation on these complicated topics.

There are some minor spelling errors- highlighted

And a couple of comments

Since oral calcimmimetics do come with GI side effects which limits compliance, without knowing the reason for compliance it will be hard to generalize ethical/economic concerns in this group of patients With start of widespread use of iv calcimimmetics, many of the above vintage patient would have been started on these medications early in their ESRD life, and this could potentially improve their life expectancy and quality.  So a real ethical question going forward would be when to stop these medications. The ESRD population in your study has already exceeded the average life expectancy of a ESRD patient.  As life expectancy improves with better cardiovascular detection/treatment and preventive measures, determining life expectancy for this group of patients will be challenging and hence make the ethical questions you raise, very difficult to answer.

This manuscript is a resubmission of an earlier submission. The following is a list of the peer review reports and author responses from that submission.

Round 1

Reviewer 1 Report

Nice concept but not sure if this will impact decision making process from clinicians' point of view. There are many treatments in today's medical world which pose the same question. In the United States, there are protocols for management of anemia/hyperparathyroidism and other aspects in ESRD patients. These are protocols are designed by dialysis run companies and nephrologists have minimal to no hold in altering it.

Author Response

Thank you for your comments.

As for the language, the paper has been reviewed by a mother tongue editor, but will also be checked again by the MDPI team.

As for the content:

* we fully agree that the issue here discussed just an example, applicable also to other drugs; this may be an advantage of the paper, and was underlined as follows:

discussion:

In this regard, the case here discussed is not exceptional and may be seen as an example of the need for integrating economic and ethical issues in the routine clinical reasoning.

*while we fully agree that economic issue, national or local guidelines often limit the clinical choices, we are lucky enough to work in a country where expensive choices of new drugs are allowed, but must be motivated. We acknowledged this in the discussion, as follows:  discussion (end):

While economic issue, national or local guidelines often limit the clinical choices, we are lucky enough to work in a setting where expensive choices of new drugs are allowed, but usually must be motivated.

with the hope that the comments have answered to your remarks, thank you for the time dedicated at improving our paper

the authors

Reviewer 2 Report

This is an original article narratively describing a monocentric experience with the advent of the recent iv calcimimetic (etelcalcetide) as an example on how clinical, economical and ethical considerations may affect the choice of this new treatment. The most original part is definitely the ETHICAL approach. Authors should expand and clarify in the discussion the ECONOMICAL impact of the conversion (lines 280-282) without the need to go to the sources. Is  etelcalcetide much more expensive than oral cinacalcet price-wise at "equivalent doses"?...In some European countries "bundling" refers to ALL DIALYSIS related meds without distinction of "oral" or "iv" route, and therefore the comparison would not be then price/cost but benefits per dialysis (unethical).  It is my opinion that the BIOCHEMCIAL CLINICAL part is the least interesting. As a matter of fact, such a small group of patients and the variability of biological data do not really add much to what we already know from important studies. They actually dilute the most interesting contribution (especially ethical decisions), and therefore authors should try to simplify the clinical part which deviates the attention from where it should really be. 

Other comments:

1) I believe that title is misleading. it should include the word iv (For example "new intravenous calcimimetic agents") since there are other calcimimetics in the pipelines (i.e. evocalcet). On the other hand I dislike the use of the words "weapons"/"problems" since it already gives a negative first impression to the reader. I would definitely avoid that since iv calcimimetics represent new problems but also new solutions.

2) Authors should avoid the use of "metabolic bone disesase". They should use the current term CKD-MBD and adapt, accordingly, the text. Renal osteodystrophy and "recently" osteopenia, osteoporosis may be a part of the complex CKD-MBD

3) Lines 55-58: An important missing effect of PTH  is that it also increases levels of calcium, phosphate and FGF-23, all of them associated with higher mortality. 

4) In the Tables (Patient column) the data-base number of the patient should not be used. Refer to them as 1-2-3...It would also be helpful to state that they have been ordered according to the start date.

5) Table 2: I do not understand the expression "no increase in alkaline phosphatase, probably a low-turnover bone disease". If this were true, the use of a calcimimetic would have been inadequate!!! 

6) Table 2: It should say Ca-P balance (Ph is incorrect and could also be confused as a typo for PTH)

7) Line 192: According to what it is written, this patient had a double reason for hypercalcemia - tuberculosis and secondary hyperparathyroidism, probably not tertiary hyperparathyrodisim-. Was etelcalcetide used to suppress PTH further, below PTH usual goals?

8) Figure 2: The words MIMPARA and PARSABIV (misspelled) should be changed to cinacalcet and etelcalcetide

9) Review lines 300-302...is it not the other way around? prevention of vascular disease has a beneficial efffect on PTH????

10) Paragraph line 304 on: I do not agree with the expression "ABSENCE OF DATA" . I know that we nephrologists do not have evidence 1A for most of our decisions (reviewed by Bover J at EDTA Budapest meeting 2019).  However, authors must admit that WE DO NOT HAVE MUCH DATA but there is not an ABSENCE as they state. As a matter of fact, we do have PROSPECTIVE data on calcimimetics and progression of cardiovascular calcification (ADVANCE study) and bone (BONAFIDE study).

I enjoyed reading your contribution 

Author Response

Thank you very much for your in depth analysis of our study.

We are now submitting a point by point answer, with the hope to acknowledge exhaustively to all the points you rose.

Comment:

This is an original article narratively describing a monocentric experience with the advent of the recent iv calcimimetic (etelcalcetide) as an example on how clinical, economical and ethical considerations may affect the choice of this new treatment. The most original part is definitely the ETHICAL approach. Authors should expand and clarify in the discussion the ECONOMICAL impact of the conversion (lines 280-282) without the need to go to the sources. Is  etelcalcetide much more expensive than oral cinacalcet price-wise at "equivalent doses"?...In some European countries "bundling" refers to ALL DIALYSIS related meds without distinction of "oral" or "iv" route, and therefore the comparison would not be then price/cost but benefits per dialysis (unethical).  It is my opinion that the BIOCHEMCIAL CLINICAL part is the least interesting. As a matter of fact, such a small group of patients and the variability of biological data do not really add much to what we already know from important studies. They actually dilute the most interesting contribution (especially ethical decisions), and therefore authors should try to simplify the clinical part which deviates the attention from where it should really be. 

Comment:

we tried to reduce the clinical part and enhance focus on the ethical one, in particular in the discussion, where we tried to make clear that the clinical issues are ancillary to the study. However, we felt that since, as clinicians, our choices rely on potential or perceived advantages, the clinical discussion was needed, in particular since the experience is still limited, and choices are often experience based .

Other comments:

1) I believe that title is misleading. it should include the word iv (For example "new intravenous calcimimetic agents") since there are other calcimimetics in the pipelines (i.e. evocalcet). On the other hand I dislike the use of the words "weapons"/"problems" since it already gives a negative first impression to the reader. I would definitely avoid that since iv calcimimetics represent new problems but also new solutions.

thank you for your comment: we changed the title as follows: 

New intravenous Calcimimetic Agents: New Options, New Problems. An Example on How Clinical, Economical and Ethical Considerations Affect Choice of Treatment.

2) Authors should avoid the use of "metabolic bone disesase". They should use the current term CKD-MBD and adapt, accordingly, the text. Renal osteodystrophy and "recently" osteopenia, osteoporosis may be a part of the complex CKD-MBD

thanks, we corrected the text accordingly.

3) Lines 55-58: An important missing effect of PTH  is that it also increases levels of calcium, phosphate and FGF-23, all of them associated with higher mortality. 

sure: we added this (we had consider this quite implicit...)

Parathyroid hormone (PTH) is a uremic toxin, high levels of which are associated not only with the classic hallmarks of bone disease, and are characterized by increase an in calcium, phosphate and FGF-23, all of them associated with an increased mortality, but also with anaemia, hypertension, neuromyopathy, dysfunction of the peripheral and central nervous systems, and various metabolic effects, including dyslipidaemia and impaired insulin secretion (10-22).

4) In the Tables (Patient column) the data-base number of the patient should not be used. Refer to them as 1-2-3...It would also be helpful to state that they have been ordered according to the start date.

thank you, you are right, we changed this in tables and figures

5) Table 2: I do not understand the expression "no increase in alkaline phosphatase, probably a low-turnover bone disease". If this were true, the use of a calcimimetic would have been inadequate!!! 

Well, we agree that this is a controversial point; indeed we noted this as a "contra"; however, the lack of stable increase in alkaline phosphatase was dissociated by the effect on phosphate, suggesting that the PTH did have some effect on bone reabsorption; furthermore, we felt that trying to lower exceedingly high PTH levels could be of interest considering the pleiotropic effect of the hormone, as described above.

We changed probe in "possible" on the account of the difficulty of metabolically characterizing bone disease without a biopsy.

We added this remark in the text:

The lack of increase in alkaline phosphatses was a controversial issue: we felt that trying to lower exceedingly high PTH levels could be of interest considering the pleiotropic effect of the hormone, as described above. The fact that lowering PTH allowed lowering phosphate levels (as described below) led us to continue the treatment, as it suggested that bone reabsorption was indeed an issue in these patients.

6) Table 2: It should say Ca-P balance (Ph is incorrect and could also be confused as a typo for PTH)

thanks, we amended it

7) Line 192: According to what it is written, this patient had a double reason for hypercalcemia - tuberculosis and secondary hyperparathyroidism, probably not tertiary hyperparathyrodisim-. Was etelcalcetide used to suppress PTH further, below PTH usual goals?

thanks for this important point: we discussed the case with our endocrinologist, that suggested that it was actually tertiary hyperPTH; we tried to condensate her history as follows:

Calcimimetic treatment was suggested by the consultant endocrinologist on the account of the need for correcting hypercalcemia (contraindications to steroids; parathyroidectomy not accepted by the patient; unsuppressed PTH levels) in the wait for response to antitubercular treatment. Antitubercular treatment allowed correction of hypercalcemia; frank hyperparathyroidism (PTH 400-800 pg/mL) became evident and the patient is now scheduled for adenomectomy.  

8) Figure 2: The words MIMPARA and PARSABIV (misspelled) should be changed to cinacalcet and etelcalcetide

thanks, we did this

9) Review lines 300-302...is it not the other way around? prevention of vascular disease has a beneficial efffect on PTH????

this was reworded as follows:

In patients with long life expectancy, prevention of vascular disease generally is crucial and PTH levels should probably be kept in the lower target range to minimize vascular damage.

10) Paragraph line 304 on: I do not agree with the expression "ABSENCE OF DATA" . I know that we nephrologists do not have evidence 1A for most of our decisions (reviewed by Bover J at EDTA Budapest meeting 2019).  However, authors must admit that WE DO NOT HAVE MUCH DATA but there is not an ABSENCE as they state. As a matter of fact, we do have PROSPECTIVE data on calcimimetics and progression of cardiovascular calcification (ADVANCE study) and bone (BONAFIDE study).

thanks: it was "absence of CLEAR" data, we corrected it as "lack of clear data"; the cited studies, in my understanding, report a different progression of calcifications, but not a reversal...

Thanks again for the time dedicated at improving our study and for the constructive comments that helped us very much. 

Reviewer 3 Report

The authors have taken a topic which has not been discussed much in Nephrology.  This can be called palliative Nephrology. I agree with the aim of the study is to raise questions and the authors have categorized the discussions well.  The discussions could be made concise.  Although there are assumptions about life expectancy and presumed benefit/or lack there of, this is a limitation of a case  series study. 

Overall a good attempt ant initiating a conversation on these important topics that will influence the treatment of elderly.

Author Response

Thank you for your kind words; in keeping with your suggestions, we have tried to shorten the discussion, and highlighted the limits of teh study.

thank you for the time dedicated at improving our study,

The authors